# Valorization of Vegetal Fibers (Hemp, Flax, Miscanthus and Bamboo) in a Fiber Reinforced Screed (FRS) Formulation

**DOI:** 10.3390/ma16062203

**Published:** 2023-03-09

**Authors:** Sergio Pons Ribera, Rabah Hamzaoui, Johan Colin, Laetitia Bessette, Marie Audouin

**Affiliations:** 1Institut de Recherche en Constructibilité, ESTP-Paris, 28 Avenue du Président Wilson, 94234 Paris, France; 2Sigma Béton & VICAT, 4 Rue Aristide Bergès, CEDEX, 38080 L’Isle-d’Abeau, France; 3Fibres Recherche Développement, Technopole de l’Aube en Champagne, Hôtel de Bureaux 2-2 Rue Gustave Eiffel-CS 90601, CEDEX 9, 10430 Rosières-prés-Troyes, France

**Keywords:** fiber reinforced screed (FRS), fiber reinforced mortar (FRM), vegetal fibers, hemp, flax, miscanthus, bamboo, cement, synthetic fibers, mortar

## Abstract

A persistent rise in the costs of construction materials has led to the need to address this problem in line with the Sustainable Development Goals. This research employed vegetal soft and rigid fibers in a screed mortar to produce a sustainable fabric–cement matrix. Four different vegetal-dried fibers (hemp, flax, miscanthus, and bamboo) with dosages of 0.4, 0.6, 0.8, 1.2, 2, and 4 kg/m^3^ were used. Laboratory investigations were slump test, bulk density, air occluded, shrinkage, and mechanical strength. Scanning Electron Microscope (SEM) assessments were performed and analyzed on the natural fibers and the screed formulation. The results highlight that fiber dosages significantly influence the above-mentioned properties.

## 1. Introduction

Concrete, with a low tensile strength, is considered a brittle material (about 1/10 of its compressive strength) [1] and has a low capacity to resist cracking under dynamic loads [2], which can create durability problems [3,4,5,6]. Mortar is also a brittle material, but, unlike concrete, it has a semi-structural nature, as it does not contain aggregates. Fiber-reinforced mortar (FRM) is so called because it contains millimeter-sized fibers that increase its resistance. The fibers in the mixture serve as a buffer against the hydraulic shrinkage that is typical of mortar during the curing process, preventing the development of cracks and fissures on the surface of the plaster that has been applied. There are many factors (internal temperature, autogenous and drying shrinkage, plastic settlement, etc.) that are responsible for the easy cracking of concrete and mortar structures, creating access for hazardous agents and, ultimately, decreasing the service life of the structure [7,8,9,10].

One of the methods to reduce concrete manufacturing costs consists of replacing synthetic fibers with vegetal fibers and reducing or optimizing the fiber content while maintaining the desired mechanical qualities, as some research has already shown [11,12].

Although it has been previously found that only by increasing the aspect ratio of the fibers it is possible to improve certain mechanical properties [13,14,15] instead of modifying the fiber dosage, this study is based on the conception of multiple fiber dosages but also takes into account the aspect ratio, due to the different lengths and diameters of each fiber, to explain the performance of each formulation.

Although it is not anticipated that mortar will increase the strength of the structural system due to its low thickness, it should be able to resist tensile stresses induced by shrinkage as well as by service and accidental loads. Similarly to concrete, mortar exhibits low tensile strength, which renders necessary the use of reinforcement to take up the tensile stresses; otherwise, cracks will appear on the screed mortar surface, jeopardizing the durability of the structure. Due to the high level of the tensile stresses that have been generated in concrete constructions at both the serviceability and the ultimate limit phases, steel reinforcement is typically added to the concrete mass. For screed mortar, however, the use of reinforcing bars is typically avoided due to space constraints, as the mortar layer is relatively thin and, in many applications, cannot ensure the prescribed reinforcement cover by the codes of practice. Moreover, the high cost of reinforcement compared to the overall mortar cost renders the use of reinforcement non-attractive [16].

For the above reasons, the use of natural or synthetic fibers for screed mortar applications has gained significant popularity in recent years. The use of synthetic fibers is, however, associated with high energy costs and a relatively heavy environmental impact compared to natural fibers [17,18].

Conversely, utilizing natural fibers extracted from plants offer significant advantages over synthetic fibers, such as significantly lower energy costs and environmental impact. Previous research has shown that vegetal fibers such as hemp, flax, bamboo, and miscanthus are a sustainable and renewable source for reinforcement in concrete [18], and their introduction in cementitious materials leads to higher concrete tensile strength [19]. The cultivation scope of the above-mentioned types of vegetal fibers in France is substantial, as the country is considered both the world’s largest flax producer, with cultivation land exceeding 106,000 hectares, and Europe’s largest hemp producer, with 17,000 hectares of hemp fields [20].

The trend of using vegetal fibers in cementitious composites is continuously growing [21], as attested by the numerous investigations conducted to develop sustainable building materials using natural fibers [22,23,24,25,26,27,28,29,30,31,32] as plant filler [24,25,26], vegetal aggregates [28], and particularly bamboo fibers [30], hemp fibers [29,32], and flax fibers [32].

Therefore, the idea of fiber-reinforced screed (FRS) has been conceived with the aim of developing a more ecological and less polluting material—more specifically, FRS with vegetal fibers, in which low cost also becomes a key factor.

This study focuses on FRS in a screed mortar formulation which, in recent decades, has been widely used as floor improvement material in industrial, water, and wastewater treatment applications because it provides considerable benefits, such as high compression strength, high abrasion resistance, and enhanced durability, which lead to longer service life and reduced maintenance and life cycle costs.

The present article focuses on screed mortar applications using vegetal fibers from hemp, flax, bamboo, and miscanthus. The novelty of vegetal fiber-reinforced screed is that it offers special advantages over traditional screed (usually incorporating fibers such as plastic, glass, or steel in its mixture) including increased sustainability, improved insulation, and enhanced durability.

An extensive experimental campaign was carried out to examine how fiber size, content, and type affected structural performance, determined through flexural and compressive strength tests, shrinkage or crack control and the workability of the mix. Moreover, Scanning Electron Microscopy was conducted to compare the surface conditions between soft and rigid fibers with mortar and gauge the range of fiber diameters, which facilitated the detailed investigation of issues related to the bond between the mortar matrix and fibers. The key results are presented and discussed in detail, while guidance is provided for engineering practice applications.

This study also highlights current challenges and the pressing need for additional study.

## 2. Materials and Methods

### 2.1. Materials

The work described herein aims to identify the key parameters that influence the mechanical behavior of screed mortar when vegetal fibers are added to the mix.

The raw fibers supplied by FRD (Fibres Recherche Développement) for usein this investigation underwent only mechanical transformation and no chemical treatment in an effort to minimize the environmental impact of the elaboration process. The focus is drawn on four types of fibers (Figure 1): (i) flax, (ii) hemp, (iii) miscanthus, and (iv) bamboo. Flax and hemp are classed as soft fibers, and their preparation involves retting, stripping (i.e., the separation of fiber from the shiver), and grinding using sieves. Every soft fiber was retted and stripped. The retting procedure is used to remove the pectin that surrounds the fiber and holds them to the stem and to one another. Between harvesting the flower and the stem, the fiber had a natural retting process in the field.

On the other hand, miscanthus and bamboo belong to the category of rigid fibers, and their preparation includes grinding and sieving. In the experiments presented and discussed in this paper, two nominal fiber lengths, *l_nom_* = 5 and 15 mm, were considered for each one of the four above-mentioned fiber types. However, the actual fiber lengths used in the investigation were smaller (as indicated in Table 1) since the nominal values are not the real average values of fiber length, even if the sieves used are 5 and 15 mm. The actual fiber length varies due to factors such as sieving, and the nominal fiber length refers to the length specified by the manufacturer. Each mix is denominated from the first letter of the fiber material followed by the nominal fiber length (e.g., F15 stands for the mix with flax and a nominal fiber length of 15 mm).

A Portland cement of type CEM II/A LL 42.5 R CP2 CE NF was used in this study, as is appropriate for screed mortar applications where high initial strengths are anticipated. The X-ray fluorescence analysis results (Table 2) indicate that the cement is principally composed of CaO (68%) and Silicium (15%).

A high purity limestone filler was added to the mixture in addition to the cement to enhance its workability. The adopted dosage for the production of the anti-fissuration screed mortar is shown in Table 3. Each mix consisted of a volume of 2 L.

All the materials used are illustrated in Figure 2.

The effective total water dosage was slightly higher than the nominal values of Table 3 (235 kg/m^3^) since the absorption of the sand, the filler, and the added dry fibers needed to be compensated for. Six different fiber percentages were evaluated in the test campaign: 0.4, 0.6, 0.8, 1.2, 2, and 4 kg/m^3^. The conversion from kg/m^3^ to fiber volume percentage (%) is provided in Table 4.

The mortars were made according to NF EN 196-1 Standard [33]. To create a homogeneous dispersion fiber in the fluid, the mixing procedure must be carried out properly. The total time of one mixing is five minutes and thirty seconds, as shown in Figure 3.

After demolding, the specimens’ conservation was upheld in accordance with standard NF P15-433 (February 1994) [34]. The samples were kept in storage at 20 °C and 95% relative humidity.

The total amount of water added to the mixture with a fiber dosage of 0.4 and 4 kg/m^3^ is shown in Table 5 (the lowest and the highest dosage rate) as a function of the fiber material and nominal length where the highest absorption coefficient of miscanthus is noted in the 4 kg/m^3^ fiber dosage.

### 2.2. Methods

All of the experiments are displayed in fresh and hardened states in Figure 4.

#### 2.2.1. Characterization in the Fresh State

##### Volumic Mass and Air Entrapped

The volumic mass was calculated according to NF EN 12350-6 [35] using the manometer receptacle (vessel) also used for the measurement of air entrapped (EN 12350-7) [36].

##### Slump Test/Consistency

This test was performed in accordance with NF EN 12350-5 [37] to assess the sample’s consistency using a little cone (Dmax = 133 mm, Dmin = 70 mm, and 120 mm height). To obtain an average result, at least four experiments were run. At T0, T60, T120, and T180 min, the diameter of the slumped mortar was measured in two directions. If the mean flow values of the two individual flow values varied by more than 10%, some experiments were dropped and replaced by others.

#### 2.2.2. Characterization in the Hardened State

##### Flexural and Compressive Strength

This test was performed following the NF EN 196-1 standard [33] to measure the flexural and compressive strengths. Three and six screed formulations were cast in 4 × 4 × 16 cm polystyrene molds for flexural and compressive tests, respectively. Tests were run on 3R (Syntech) 300 kN test equipment at intervals of curing ages of 1, 7, 28, and 90 days.

##### Shrinkage Test

This test was run in compliance with NF P 18-427 [38] to find out how long the screed sample would take to crack. The measurements were made with the use of a specialized tool (shrinkage meter) that included a comparator.

The samples were stored in a controlled climate room (at a temperature of 20 ± 2 °C and a relative humidity of 50 ± 5%) for the duration of the test. At the beginning, shrinkage measurements were made on three specimens, each measuring 4 × 4 × 16 cm, of each formulation at extremely brief intervals. With time, the periodicity of the measurement grew (at 1, 3, 7, 14, 21 and 28 days).

##### Scanning Electron Microscopy (SEM)

The raw fibers (hemp, bamboo, miscanthus, and flax) and the fibers in the cement mortar were examined by SEM in order to better understand the interaction between the fibers and the matrix as well as the behavior of the vegetal fibers in the cementitious matrix. The SEM used (GEMINI model) was coupled to a Zeiss SUPRA 55VP energy dispersive spectroscopy (EDS) probe. This probe gave a more precise qualitative and semi-quantitative indication of the basic constituents of the samples being tested. In order to make the samples visible by SEM, they were first put on self-adhesive plates and then metallized in a High-Resolution Carbon Evaporator (Leica EM SCD500).

## 3. Results and Discussions

### 3.1. Characterization in the Fresh State

One of the important parameters discussed throughout this article is the aspect ratio (AR: length/diameter); it is thus presented in Table 6. The AR was calculated by taking into account the actual length of the fibers (Table 1) and the average diameter calculated from the SEM images taken for each fiber in its raw state and with the fiber embedded in the mortar.

The great difference between the AR of the soft fibers is because there was a transformation from a bundle of fibers to a single fiber between the raw fibers and the fibers embedded in the mortar, increasing the AR in the latter of these cases due to degradation resulting from the cementitious environment. This transition affected the aspect ratio of the fibers, which affected their properties in the fresh and hardened states. Since it is more representative of the impact of the fibers embedded in the mortar, the aspect ratio provided in the right column of Table 6 was considered for the analysis of the performance of the FRS, where the higher AR of the soft fibers compared to the rigid ones was noticed.

#### 3.1.1. Bulk Density (Volumic Mass)

The density of the mortars under study is depicted in Figure 5 as findings.

It is clear that adding natural fibers to a mortar causes the material’s density to drop. This quality is crucial since it makes no sense to make a given building material resistant if it is very heavy [39]. It is a good idea to lighten mortars by utilizing natural fibers because doing so significantly reduces the structure’s dead weight [40]. On the contrary, it is desirable that mortars have a higher density of mass in the fresh state. It is known that mortars with a higher density of mass in the fresh state have a higher standard of performance in application and execution, which leads to a higher saving of material; thus, a balance must be found between fiber addition and density loss.

The measured values of the mortar density are listed in Table 7 and illustrated in Figure 5.

It can be observed that the bulk density of the reinforced mortars gradually decreased when the dosage of fibers was increased. This decrease compared to the control sample, ranging from 0.40% (B5 0.4 kg/m^3^ of fibers) to 12.02% (H15 4 kg/m^3^ of fibers), was caused by the fibers’ alveolar structure and the increased porosity from adding fibers. Given that the fibers’ density was lower than that of the mortar, this was to be expected. The creation of voids at the interfaces between the fibers and the solid matrix as a result of air bubbles being trapped by fibers during the mixing process can also be used to explain these findings [41]. Due to the large porosity of the resulting material’s fibers, this reduction is also largely attributable to them [42]. Equivalent results were found by Siham Sakami et al. [41], in which the bulk density decreased by up to 14.68% for 5% of alfa mass content when using a reinforcement of alfa fibers.

Another conclusion that can be drawn from the graph (Figure 5) is that soft fibers embedded in mortar have a much more significant drop in volumetric mass than rigid fibers, especially at maximum dosage.

This can be explained by the fact that the AR is much higher for soft fibers. This higher AR means that the fibers are slenderer, so more fibers are required for the same dosage, so that, as the number of fibers increases, there are fewer aggregates and the density decreases. Another immediate consequence of a higher AR is a greater loss of workability and a major increase in occluded air with soft fibers compared to with rigid fibers. On the other hand, there should be a greater shrinkage control with soft fibers; however, in the shrinkage section it can be seen that the difference is not remarkable: although the flax FRS had the least shrinkage, the bamboo FRS had a minimal shrinkage difference compared to this one.

Evidence can be gathered through the analysis of Figure 5 and Table 7 to identify whether the low density of the mortar is caused by the inclusion of the fibers or the development of pores. When the density of the fiber-reinforced screed was compared to that of a control sample free of fibers, it was observed that the density of the soft FRS with a greater dose of 4 kg/m^3^ was lower. This suggests that the fibers were likely to blame for the low density. Although the density of the control sample was always higher, the other FRS densities were pretty close to the latest, suggesting that the creation of pores may be the main reason for the low density except in the case of high dosages of the soft fibers.

It is preferable for mortars to have a larger mass density when they are fresh. It is well known that mortars with higher mass densities when they are fresh perform better when placed and used, which leads to a higher saving of material. In turn, it can be concluded that the incorporation of fibers is beneficial.

#### 3.1.2. Control of Air Content

Figure 6 and Table 8 show that the values of air entrapped increased with an increase in the quantities of the vegetal fibers.

Additionally, it has been shown that the soft fibers (flax and hemp) had higher values of entrapped air than the rigid fibers (miscanthus and bamboo). Thus, the structural arrangement of the fiber strands and lower densities than the cement led to higher values of entrapped air. The vegetal reinforced composites had lower unit weights than that of the control mortar due to the increase in air content and the lower weights of the fibers.

The air content of vegetal fiber mortar composites depends on the fiber volume, workability, and mixing methods, all of which influence air dispersion from fresh mortar samples [39]; as we can see in Figure 6, flax and hemp had the highest values of air entrapped with the biggest dosage of 4 kg/m^3^, corresponding to a lower workability and difficulties with the soft fibers during the mixing because they were more abundant and slender (AR ratio higher than the rigid fibers), which meant that the fibers agglutinated and clumped into each other.

H15 with a fiber dosage of 4 kg/m^3^ had the highest entrapped air in the fiber composite, with a value of about 5.4%, followed by H5, with 5.2%. Hemp fibers have a lower density due to the structural arrangement of the fiber; hemp is believed to be porous, with a tendency to entrap more air than others. Although it is not the only factor, moisture absorption contributes to the porosity of hemp fibers. Hemp fibers expand when they take in moisture, widening the spaces between the filaments. The fiber’s general porosity rises as a result of this. According to studies, hemp fibers had an equilibrium moisture content (EMC) of 11.6% [43], flax fibers were roughly 8% [44], miscanthus fibers were 9% [45], and bamboo fibers were 6% [46] at 25 °C and 65% RH. The greater the quantity of entrapped air in a composite the lighter it becomes, which is corroborated by the bulk density results in Figure 5.

Porosity is referred to as the number of voids by volume in porous material; it is a measure of the volume of air per unit volume of the said material, expressed as a percentage. Furthermore, the addition of fiber to mortar decreases the density and increases porosity [47,48]. This accounted for the measurements of density for the fiber composites and the control samples.

The addition of fiber increases porosity [49] and therefore moisture absorption, which subsequently reduces the compressive resistance of the fiber in the composite [50].

The air content test results shown in Figure 6 reveal that, when the fiber dosage was minimal, there was only a very minor impact on the number of pores in the mortar mix. The number of pores grew as the number of fibers in the mortar increased. Similar regularity was observed by Vafaei et al. [51], the authors concluding that this was caused by air voids trapped in the fresh mixture due to problems with fiber distribution and orientation. In this study, the highest increase in air entrapped (Table 8) was observed for H5 samples with 4 kg/m^3^ of fibers (80% compared to CS), followed by F5 samples with the highest fiber dosage (70% compared to CS); thus, as in the case of volumic mass, soft fibers have a major impact on the occluded air in the cementitious matrix.

It is to be noted that convincing evidence was not found for variations in either volumic mass or occluded air in fibers of nominal length from 5 mm to 15 mm. Accordingly, fibers with a lower AR ratio, such as 5 mm fibers, should perform more effectively due to less occluded air and therefore a higher density, but that is not the case for every dosage.

Regarding the occluded air depicted in Figure 6, it appears that all fibers had more air than before. These findings suggest the decreasing workability of mortars containing fibers because the material’s workability decreases with increasing air content. Thus, it can be noted that the obtained values of incorporated air are coherent and advantageous in terms of the workability of the mortars because a significant increase in occluded air was not observed until 4 kg/m^3^ was reached. Since the air content is directly related to the workability of the mortars, the trend observed proves to be consistent and can be justified by the same reasons given in explaining the workability of the mortars [52,53,54,55,56,57,58].

#### 3.1.3. Slump Test/Consistency

The slump value of cement composites is an essential factor in measuring the workability of a fiber cement mortar, in which the following factors take an active role: the water–cement ratio, properties of the material, mixing methods, dosages, and admixtures.

The flow results to T0 of reinforced mortars with vegetal fibers in the amounts of 0.4, 0.6, 0.8, 1.2, 2 and 4 kg/m^3^ are illustrated in Figure 7.

The references of the control specimen and with synthetic fibers with a dosage of 0.6 kg/m^3^ are included. This reference dosage has been chosen because it is the commercialized dosage by Vicat with which all the results are compared. It should be noted that the flow of the reinforced mortars dropped as the fiber content rose beginning at 0.8 kg/m^3^. This decrease can be neglected at the scale of the rheological characteristic of the mortar except for the dosage of 4 kg/m^3^ in the case of the soft fibers, where the loss of workability became appreciable. Due to the interlocking of the fibers, it is obvious that increasing the fiber dosage and aspect ratio (AR) has a negative effect on workability. This result is in accordance with several studies [52,53,54,55,56,57,58].

All values found for both the soft and rigid fiber formulations had acceptable flow rates with no segregation. The slump values remained stable until 1.2 kg/m^3^, when the slump started to decrease drastically for the soft fibers, while for the rigid fibers the slump remained almost the same.

The main parameter that can explain this behavior is the aspect ratio AR (length/diameter) of the fibers (parameter set in Table 6). For the soft fibers embedded in the mortar, this ratio was higher than for the rigid fibers, making them slenderer, which created more blending problems with the consequent loss of fluidity and the lower slump value.

The dosage of 1.2 kg/m^3^ was the upper limit dosage beyond which the results did not improve with the addition of extra soft fibers, tending to fiber balling. Through that statement, it can be observed below in the mechanical strength section that the greatest results at 90 days of cure with soft fibers were obtained with a dosage of 1.2 kg/m^3^, which could be justified by the negative effect that the soft fibers had on the mortar’s mechanical strength when agglutinated in higher dosages.

As the results up to a dosage of 0.8 kg/m^3^ were very close at 0 min, tests were carried out over time (at 60, 120 and 180 min) to see how the time factor influences the decrease in the slump value for the mortars with 0.4, 0.6, and 0.8 kg/m^3^ fiber dosages. A range between 380 and 420 mm in diameter was considered a satisfactory slump result taking into account the values proposed by DTA N° 13/18-1387-V1 of CSTB.

Several reports [32,59,60] have verified the samples reinforced fibers’ low workability, but, in this work, we could not confirm this in the case of the rigid fibers, which presented good behavior even in high dosages.

The aspect ratio, geometry, volume fraction, matrix proportions and fiber–matrix interfacial bond qualities of FRS in its freshly mixed form all affect its attributes [61]. Good workability should be ensured for conventionally placed FRS applications to allow placement, consolidation, and finishing with the least amount of effort while providing uniform fiber distribution and preventing segregation and bleeding. As with ordinary concrete, the degree of consolidation affects the strength and other qualities of the hardened material for a particular composition. The addition of fibers may lessen the composite’s observed slump as compared to a non-fibrous mixture in the normal volume fraction ranges utilized for FRS, which is between 25 and 102 mm [40], as can be evidenced by the results obtained for every slump (Figure 7).

A mixture with a low slump can have excellent consolidation capabilities, according to studies [62]. FRS and non-fibrous mortar exhibit similar time-dependent slump reduction properties [63]. In addition to the aforementioned factors, it is important to prevent fiber balling. When shaken together, soft fibers such as hemp and flax, which have an aspect ratio greater than 100, will tend to interlock to produce a mat or ball that is very problematic to separate. On the other hand, rigid fibers such as miscanthus and bamboo, which have an aspect ratio of less than 50, are unable to interlock and are easily scattered [39]; therefore, the slump of soft fibers is much less fluid than that of rigid fibers for all lengths and dosages (Figure 7).

The maximum size and overall gradation of the aggregates used in the mix, the aspect ratio of the fibers, the volume fraction, the fiber shape, and the method of introducing the fibers into the mix are all factors that affect an FRS mix’s propensity to clump or produce fiber balling in the freshly mixed state. The lowerst volume percentage of fibers that can be added without a tendency to clump or ball depends on the maximum aggregate size and aspect ratio. Figure 7 represents slump test values for the soft and rigid fibers at 0 min, while Figure 8a–d illustrates slump tests over time (at 0, 60, 120, and 180 min) for the two soft and two rigid fibers separately. It can be seen from the charts that higher dosages of the soft and rigid fibers decreased the workability of the formulation [52,53,54,55,56,57,58] because the addition of fibers led to an increase in the surface area, affecting the workability.

The big decrease in flow at a soft fiber dosage of 4 kg/m^3^ (Figure 7) was due to the phenomenon of fiber agglomeration, which is difficult to avoid at a high reinforcement ratio. The presence of small fiber pellets in the mortar slurry resulted in a lower flowability of the mortar compared to normal mortar, which, combined with a high AR, resulted in a lower flow result.

Workability increases with an increase in moisture content [41]. During the first and final setting times, the rheological characteristics of fresh mortar change constantly, which leads to a reduction in workability and increased energy consumption during subsequent consolidation [42].

A higher AR ratio resulted in a smaller flow, so it was logical that the use of shorter fiber lengths should enhance the slump; this was the case for most fibers in all formulations, although the minor differences are not significant. Instead, the higher dosage of fibers did lead to a loss of workability in all formulations.

In Figure 7, the flow at 0 min is represented and it is observed that, for all dosages, the flow was lower than the sample control and higher than the formulation with a dosage of 0.6 kg/m^3^ of synthetic fibers, the exception being the soft FRS with the highest dosage of fibers, 4 kg/m^3^, which decreased up to 325 mm in the case of F15.

In Figure 8a–d, flow over time is represented, showing that all formulations were less fluid than the control sample. On the other hand, the formulation with synthetic fibers performed very similarly to those with the vegetal fibers. Considering that a good slump range is between 380 and 420 mm (depicted by the two black horizontal lines in Figure 8), the slumps from 60 to 120 min fit better in this range for all the formulations. A more fluid slump was observed for all formulations with the rigid fibers compared to those with the soft fibers; this was due to their lower AR ratio, which makes rigid fibers less crimped with the mix and more workable.

The increment in fiber content increases water absorption, reducing the water-cement ratio with a resultant decrease in the workability. The water absorption of a mix is a vital property because it influences its water–cement content and other properties.

The volumetric content of the fiber is one of the most critical parameters that adversely lowers the workability of the screed mortar [52]. The higher the W/C (water/cement) ratio, the higher the concrete’s workability.

Looking at the trend in the workability (Figure 8a,b) values from H5, H15 and F5, F15 at 120 min, there was a loss of workability as the fiber dosage increased. Hence, these findings strongly agree with those in the literature that workability decreases with an increase in the fiber dosage.

From Figure 8c,d, it can be seen that the bamboo had higher slump values than the miscanthus, giving it the highest slump of all the formulations. However, B15-0.8 with a higher dosage had more fluid behavior; therefore, the fiber type had a greater impact on the slump result than the fiber dosage in the case of the rigid fibers below a dosage of 1.2 kg/m^3^.

### 3.2. Characterization in the Hardened State

#### 3.2.1. Flexural Strength

Flexural and compressive tests at 1, 7, 28 and 90 days were performed, but 28 and 90 days were considered sufficiently representative.

Figure 9 and Figure 10a–d represent the flexural (Figure 9) and compressive (Figure 10) strength of the soft and rigid fibers, respectively, with varying contents of 0.4, 0.6, 0.8, 1.2, 2, and 4 kg/m^3^ at 28 and 90 days of curing age. The flexural and compressive strength results of every mortar formulation were compared with the sample control and the formulation with synthetic fibers, which was also taken as a reference for comparison with every mortar formulation in Table 9 and Table 10a,b for flexural and compressive strength, respectively.

The results presented in Table 9a show that the incorporation of fiber in the mortar increased the mortar’s flexural strength in the order of 20–30% compared to the antifissuration screed (AS) mortar, reaching up to 57.77% at 28 days, even with negative percentage values for 4 kg/m^3^ of the soft fibers. In the case of the rigid-fiber FRS, the flexural strength was in the order of 20–30% more than AS mortar, attaining 50.33% at 28 days of cure. The strength at 28 days was more significant at a small dosage of 0.4 kg/m^3^ for the soft fibers, while for rigid fibers is for 2 kg/m^3^, the same fiber dosage as for 90 days (Table 9b). Otherwise, where the mortar strength reached its maximum value at 90 days for the soft fibers was for a fiber dosage of 1.2 kg/m^3^, increasing up to 70.79% in the case of H5.

It is clear that fiber dosage and aspect ratio have a positive influence on flexural strength [56,64,65], so the behavior of the soft fibers against the literature may be explained by the high loss of bulk density with a dosage of 4 kg/m^3^, which is strongly correlated with the high AR of the soft fibers entailing more and slenderer fibers, and therefore fewer aggregates that decrease the mechanical strength. Other immediate consequence of the higher AR were the greater loss of workability, the major increase in occluded air, and the lower mechanical resistance compared to the rigid fibers.

On the other hand, the rigid fibers, which had a lower AR, showed a logical response in which higher flexural strengths were found with higher fiber dosages.

Another reason that may explain this anomalous behavior by the soft fibers is that, although increasing the fiber dosage generally increases fiber load bearing efficiency, if the AR is too high the fibers may become tangled during mixing, resulting in poor fiber dispersion which can reduce the overall reinforcement efficiency [66,67,68,69].

Analysis of the slump test shows that the 1.2 kg/m^3^ dosage represents a turning point from which the slump started to decrease strongly for the soft fibers, and, since workability is linked by the AR to mechanical strength, it can be concluded that the optimum dosage for the soft fibers is 1.2 kg/m^3^ in terms of workability, flexural strength, volumetric mass, and occluded air; although the latter two parameters did not have the best values, they had average values not far from the optimum ones.

Regarding the soft fibers, a consistent superiority in terms of the flexural strength of hemp over flax was noted; as D. Sedan et al. (2007) cited in their paper, hemp fiber composites have a high suppleness as well as a higher flexural strength than cement by itself [70].

Soft fibers composites (hemp and flax) have lower flexural strength values than the rigid fibers (miscanthus and bamboo) irrespective of the length of the fiber, except for H5 with 0.4 and 1.2 kg/m^3^ of fiber dosage at 28 and 90 days respectively.

On average, bamboo has the highest flexural strength due to its properties which enhance said strength [71] over miscanthus and the soft fibers, with the above-mentioned exception of H5.

Kriker A et al. [64] concluded that increasing fiber lengths improves the flexural strength and toughness of the composite. This makes sense because flexural strength is a longitudinally distributed force, and the longer the fiber the more it contributes to increasing the flexural strength; however, this is not confirmed by the results obtained in this study either for the soft or for the rigid fibers.

It can be evidenced that all formulations at different ages outperformed the formulation with synthetic fibers, but not the control sample.

#### 3.2.2. Compressive Strength

The results presented in Table 10a show that the highest increase in compressive strength was observed with the lowest fiber dosage for both the soft and rigid fibers at either 28 or 90 days of curing age. Therefore, these results are in line with those in the literature stating that the higher the fiber dosage the lower the compressive strength [64,72].

In all formulations and at every age, the 0.4 kg/m^3^ fiber dosage had a good performance in terms of compressive strength. Although a decrease in compressive strength was observed as the fiber dosage increased, for a dosage of 1.2 kg/m^3^ better results were found for both the soft and the rigid fibers at either 28 or 90 days of curing age.

The incorporation of fiber in the mortar increased its compressive strength in the order of 20–40% (up to 43.53%) compared to the antifissuration (AS) screed mortar at 28 days, with a negative percentage for 4 kg/m^3^ soft fiber dosage in the case of H15 and in the order of 30–60% (up to 65.59%) at 28 days for rigid fibers.

The trend at 90 days of curation was similar to that at 28 days, increasing the compressive strength up to 46.50% compared to the AS mortar for 0.4 kg/m^3^ and up to 59.95% for 1.2 kg/m^3^ of soft fiber dosage. Furthermore, in the case of the rigid fibers the compressive strength rose up to 60.45% and 66.65% against the AS mortar for 0.4 and 1.2 kg/m^3^ fiber dosage, respectively, both for bamboo; thus, as for flexural strength, bamboo had the highest increase in terms of compressive strength. On the contrary, for the highest fiber dosage of 4 kg/m^3^ it was observed even in negative percentages in the case of H15, meaning that its compressive strength was lower than that of the synthetic fiber formulation. Except for these results, however, all formulations at 28 and 90 days of curation age exceeded the compressive strength value of the synthetic fiber formulation and, for certain dosages of rigid fibers and H5, the control sample as well.

According to Page, J. and Kriker A et al., the greater the length of a fiber the lower its compressive strength [64,72]. This performance was observed in most of the formulations, but to a very minor degree, which does not corroborate the aforementioned.

As for the superior compressive strength of the rigid fibers compared to the soft fibers, it is worth mentioning the AR coefficient: soft fibers with a higher AR become entrapped during the mix, impairing their dispersion and intermixing, with a considerable loss of performance [66,67,68,69].

Overall, the compressive strength results are in consistent accordance with the bulk density and occluded air results. As observed in Figure 5 and Figure 6, bulk density and occluded air showed a corresponding performance with that of compressive strength: a lower bulk density and a higher amount of occluded air led to a consequently lower compressive strength, a fact that can be evidenced by the lower strength achieved with the higher fiber dosage of 4 kg/m^3^ (Figure 9a–d).

#### 3.2.3. Shrinkage Test

One of the most frequent causes of pavement cracking is shrinkage cracking. Reinforcing the mortar using short, randomly placed synthetic fibers is one way to lessen the detrimental impacts of shrinkage cracking; vegetal fibers were also tested in this experiment to compare their effects.

Since screed mortar is frequently under tension, cracking is a regular occurrence. Fibers serve three purposes in these circumstances: the composite retains its residual tensile strength even when shrinkage cracks appear because (1) they permit multiple cracks to form, (2) they permit tensile stresses to be transferred across the cracks, and (3) stress transfer can take place over an extended period of time, allowing the cracks to heal or close [73].

Measurements of withdrawals started 24 h after the samples were manufactured and then at 7, 14, 21, and 28 days.

The comparator had been calibrated prior to each measurement series. The curves show that each fiber had a behavior that differed from the others.

Certain test results [74,75,76] show that the creep and free shrinkage behavior of Portland cement mortar are unaffected by fiber reinforcing in modest quantities; this is why a formulation with 2 kg/m^3^ of fiber dosage is produced.

The evaluation of the drying shrinkage as a function of the length and type of the fibers is shown in Figure 11. All fibers are represented, both soft and rigid, as well as the control sample and the formulation with synthetic fibers.

Up to 14 days, the shrinking increased gradually but steadily as a result of deformation caused by the drying of the mortar sample resulting from hygrometric variations in the environment [65]. Then, the shrinkage gradually increased until the test’s 28-day ending.

The least efficient FRS decreased the drying shrinkage by 1.01% (F15) compared to the antifissuration screed (AS) mortar, while the most efficient decreased it by 10.73% (F5). The other formulations did not present a reduction in shrinkage with regards to the formulation using synthetic fibers.

From the results, it can be noted that all results are within the target of 800 µm/m (described in CSTB Report No. 3774) and that, throughout the days, they all showed a progressive shrinking, M5 and M15 being the formulations with the highest shrinkage with 623 and 670 µm/m, respectively.

Additionally, compared to hemp, the reported mixtures based on flax aggregates demonstrated lesser shrinkage deformations [57], the same as bamboo with respect to miscanthus.

Evaporation and absorption are two ways that fresh screed mortar loses water, which causes plastic shrinkage. In this case, we focused on absorption by fibers. Such water loss can make the impacts of surface evaporation worse. It is widely acknowledged that the shrinkage of mortar results from a loss of water from the paste component owing to external forces, which creates negative capillary pressures and causes the paste’s volume to contract, hence the shrinkage.

The water mobilization for vegetal particles will therefore be greater in FRS with miscanthus due to its high absorption coefficient; therefore, the loss of water produced by increasing the number of fibers may cause a shrinkage increase [77,78,79,80].

If the aspect ratio is taken into consideration in addition to the absorption coefficient of the fiber, to reduce the shrinkage of composites, high-aspect ratio fibers perform better than low-aspect ratio fibers [81].

According to Table 6, it is evident that the ARs of the soft fibers once immersed in the matrix were much higher than those of the rigid fibers due to their smaller diameter. This yielded a higher shrinkage control and, therefore, a lower shrinkage, which can be evidenced in Figure 9; the majority of the soft-fiber mortar formulations had a lower shrinkage than the rigid-fiber ones.

We note that flax fiber is able to reduce shrinkage by up to 10.73% more than synthetic antifissuration fiber. This could be due to its rough surface and the matrix penetrating the fiber, ensuring suitable matrix adhesion/fibers and allowing control of the crack opening, resulting in decreased shrinkage values [80].

Despite the fact that the presence of fibers may not clearly diminish the overall amount of restricted shrinkage, they can increase the number of cracks and hence decrease the average crack widths. It is established that even a minimal number of fibers can reduce the shrinkage, as in the case of the flax formulations [65].

#### 3.2.4. Scanning Electron Microscope (SEM)

The microstructures of the raw fibers and the fibers mixed into the cement mortar at 28 curing days are shown in Figure 12a–h.

These SEM images show the distribution of soft and rigid fibers in the mortar. These micrographs provide a detailed examination of the interaction between the matrix and the fibers. All of the fibers are seen to have a rough surface, which helps them adhere well to the cementitious matrix. 

As for the hemp fibers, they are actually bundles of hemp fibers, as can be appreciated in Figure 12a, which, when they enter the mortar, become unraveled and defilagrated and completely transform from a straight fiber bundle with a diameter of 0.194 mm to a fiber with many folds and a much smaller diameter of 0.013 mm (Figure 12b). This can be explained by the fact that Ca(OH)_2_ (portlandite dissolved out of the cement material) splits the fibers into filaments by reacting with the hydroxyl groups of cellulose and hemicellulose, which contributes to the structure’s disintegration. This fibrillation broadens the contact and adhesion surface area with the matrix [82]. Another explanation could be due to the mixing process, which could destructure the fiber bundles.

The same phenomenon occurs in the case of flax, which goes from a bundle of fibers with a diameter of 0.054 mm (Figure 12c) to a much finer fiber and bent with a diameter of 0.012 mm (Figure 12d).

This significant transformation of the soft fibers impacts the AR, increasing it substantially and therefore impairing all the properties related to this ratio in both the fresh and hardened states.

On the contrary, rigid fibers maintain their shape and diameter once introduced into the matrix, as can be noted in the case of bamboo (Figure 12e,f) and miscanthus (Figure 12g,h), which means that the AR remains constant and hence much lower than the AR of FRSs with soft fibers.

The soft fibers embedded in the mortar are very similar to synthetic microfibers, with a very high slenderness that turns them into efficient fibers with very good bonding to the matrix. This means that, when a crack appears, the fiber will act as an anchor and allows a lot of deformation before the total rupture of the specimen. Furthermore, as they have a high AR, they have a good plastic shrinkage control, reducing the appearance of bleeding. On the other hand, being thinner fibers, they are also more abundant, and this generates workability problems: they have a tendency to tangle and generate fiber balling, and slumps are much thicker than with rigid fibers even in short intervals of time, e.g., at T0.

Rigid fibers with diameters ranging from 0.2–0.3 mm cannot be considered microfibers but macrofibers. In general, the fibers did not increase the mechanical strength of the FRS specimens, but the better performance of the FRSs with rigid fibers in terms of mechanical strength can be explained by the fact that rigid fibers have a lower AR, are straight fibers without bends, are more ductile, have a greater cross-section per surface, and, in particular, have a stronger containing effect.

## 4. Conclusions

The use of vegetal fibers to reinforce mortar yields to the modification of some characteristics of the FRS:An increase in fiber dosage and aspect ratio (AR) negatively impacts workability. Soft fibers start to have a non-fluid flow behavior from 1.2 kg/m^3^ of fiber dosage, whereas rigid fibers remain smooth at all dosages (at T0). The highest spread value was noted for Miscanthus 5 at a fiber dosage of 4 kg/m^3^, more than 100 mm higher than Flax 15, which had the lowest spread. FRS loses workability from a fiber dosage of 1.2 kg/m^3^, particularly with soft fibers, tending toward fiber balling. All the results obtained in the case of natural fibers showed a better spreading behavior than synthetic fibers, with the exception of the soft fibers for a dosage of 4 kg/m^3^. All values found for both soft- and rigid-fiber formulations had no segregation.The use of natural fibers in mortar leads to a decrease in the bulk density, much more significant in the soft fibers than in the rigid fibers for higher fiber dosages. The highest density decrease (about 12%) was observed in the case of Hemp 15 compared to the control sample (containing no fibers) and this for dosages of 4 kg/m^3^ in the soft fibers. This result was compared to that obtained in the case of rigid fibers such as Miscanthus 15, where the most significant decrease in density was 3% for the same dosage. Furthermore, this decrease in density observed in both cases was induced by a greater increase in the occluded air quantity for both soft and rigid fibers: 80% in the case of H15 and 53% in the case of M15. Except in the case of high dosages of soft fibers, the presence of pores may be the primary cause of the lower density of the FRS compared to the control sample. An increase in occluded air is noted with a higher fiber dosage, the occluded air being bigger in the case of soft fibers. It has been determined that the properties of bulk density and air content are complementary, meaning that the addition of natural fibers to the mortar causes a decrease in bulk density and an increase in air content. The greatest decreases in density and increases in air occluded occur in the case of soft fibers at high dosages.Flexural strength (FS) is positively influenced by the fiber dosage, bamboo and hemp being the best-performing fibers. All values were higher than that of synthetic fibers FRSs. In terms of the dosage of fibers, two points were noted: an increase in FS at 90 days of curing of 67% compared to the antifissuration screed (AS) mortar for a dosage of 2 kg/m^3^ in the case of Bamboo 5; and a 71% increase compared to the AS mortar for a dosage of 1.2 kg/m^3^ in the case of Hemp 5. The mechanical performance of these two formulations exceeded that of the control sample.The best results in terms of compressive strength (CS) were found with lower fiber dosages and 1.2 kg/m^3^ for both soft and rigid fibers. All CS values were higher than that of the synthetic-fiber formulation, bamboo being slightly superior. Rigid-fiber formulations have higher CS than soft-fiber ones due to their lower AR. Although the dosage of 0.4 kg/m^3^ performed adequately, the optimum dosage is 1.2 kg/m^3^ for all fibers and at all ages of curation. H5 achieved a 60% increase in CS over the AS mortar at 90 days, while B5 resulted in a 67% increase in CS over the AS mortar at 1.2 kg/m^3^. Both H5 and B5 performed well in both CS and FS and exceeded the control sample value.All the formulations conform to the target of 800 μm/m (0.8 mm/m), and soft fibers (higher aspect ratio) have a better shrinkage control than rigid ones, with flax standing out. Similar performances by the control sample and FRS with synthetic fibers have been gathered.All of the fibers appear to adhere well to the matrix, according to SEM observations, which is consistent with the increase in mechanical strength. Soft fibers alter their shape as they are mixed with the cement, changing the AR ratio from their original state.

This paper has reviewed the research that has focused on improving slump, entrapped air, and bulk density in the fresh state, and flexural strength, compressive strength, and SEM images in the hardened state, including the effect of fiber aspect ratio and fiber dosage. FRSs now compare favorably with plain mortar and FRSs with synthetic fibers in terms of cost and rigidness; the values of FRSs are approaching and at times exceeding those for plain mortar but always improving those for FRSs with synthetic fibers. FRS have a remarkably wide range of uses, including outdoor ones such as screed flooring. To broaden their applicability, which would entail increasing fiber ageing and long-term durability, more research is still needed. Overall, the uptake of FRS is expanding quickly, and it would seem that its application has a bright future.

## Figures and Tables

**Figure 1 materials-16-02203-f001:**
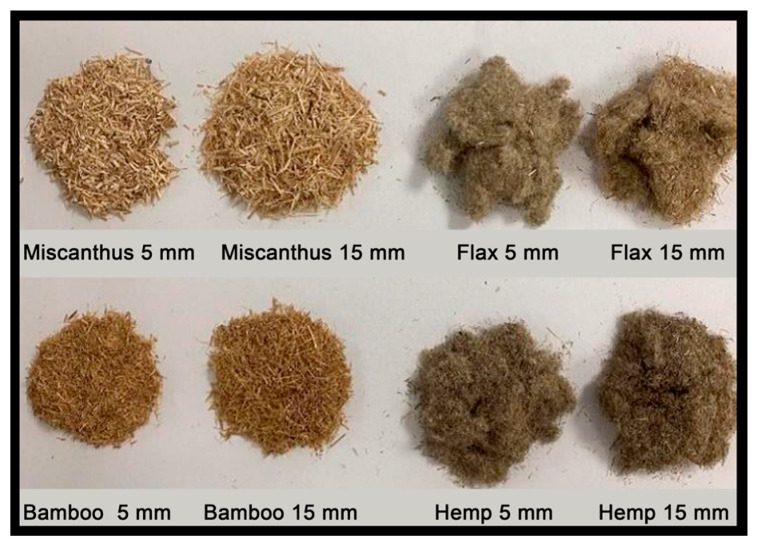
Fibers used in the research.

**Figure 2 materials-16-02203-f002:**
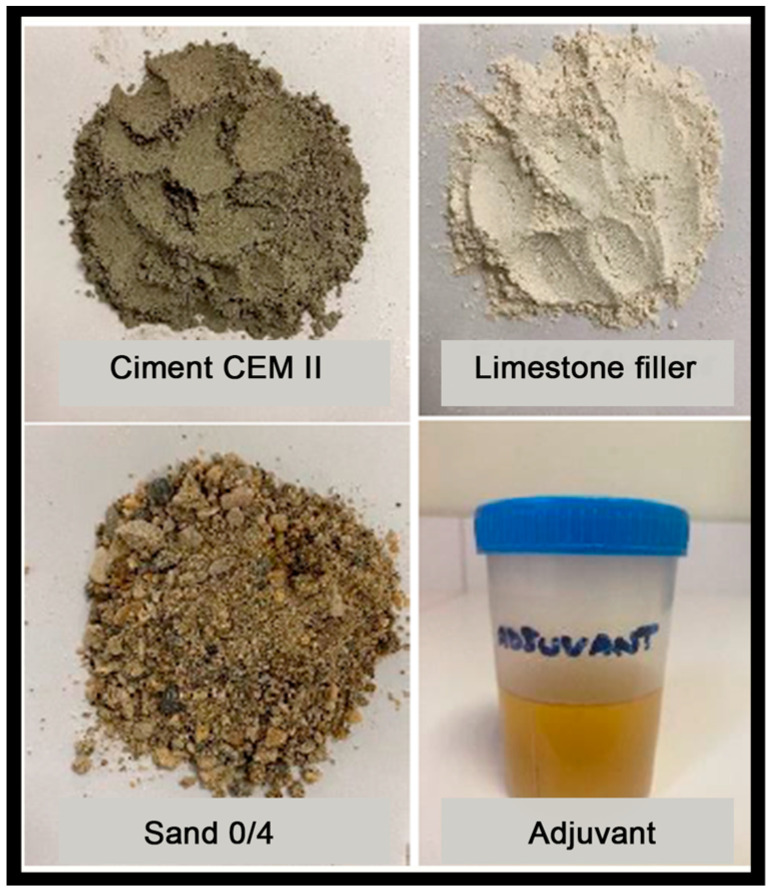
Materials used in the research.

**Figure 3 materials-16-02203-f003:**
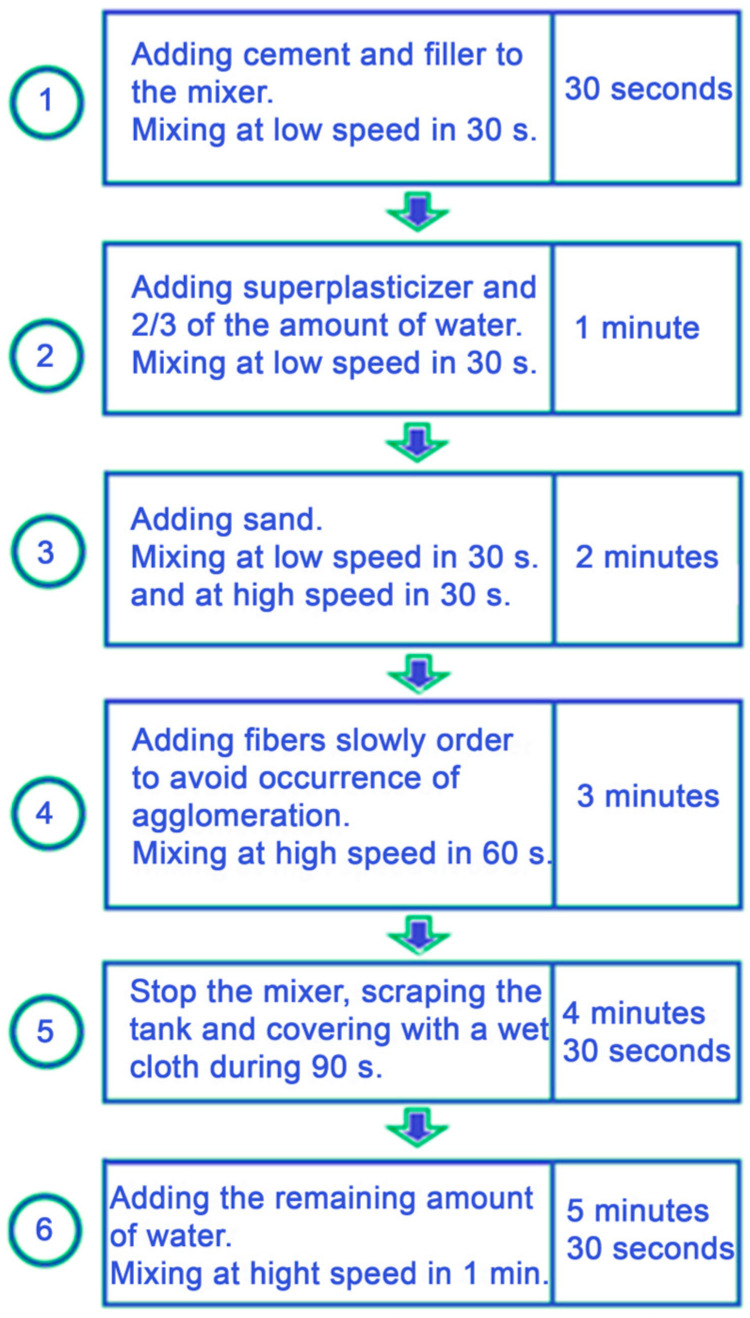
Mixing FRS process.

**Figure 4 materials-16-02203-f004:**
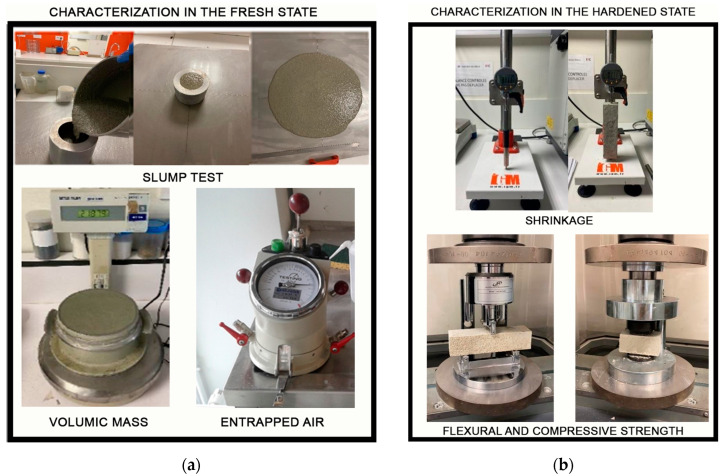
(**a**) Characterization in the fresh state. Tests carried out: slump test, volumic mass, and entrapped air. (**b**) Characterization in the hardened state. Tests carried out: shrinkage, flexural, and compressive strength.

**Figure 5 materials-16-02203-f005:**
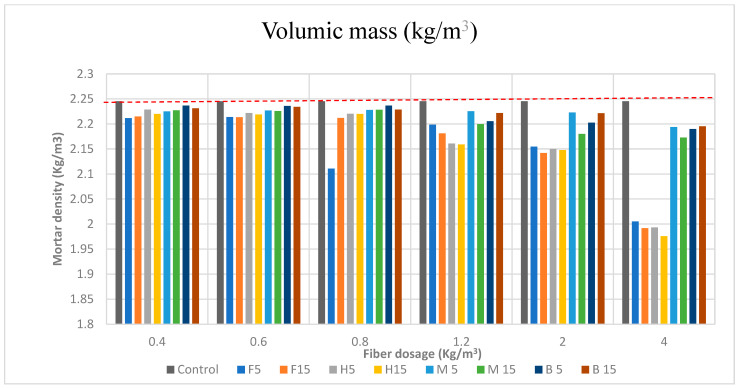
Mortar density (kg/m^3^) for every fiber dosage. The sample control density is indicated by a dashed red line.

**Figure 6 materials-16-02203-f006:**
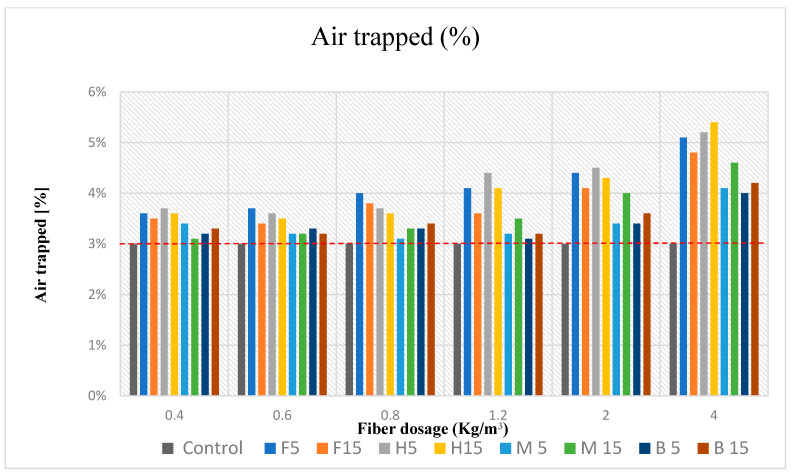
Entrapped air for every dosage of formulations for soft fibers and rigid fibers. The sample control entrapped air is indicated by a dashed red line.

**Figure 7 materials-16-02203-f007:**
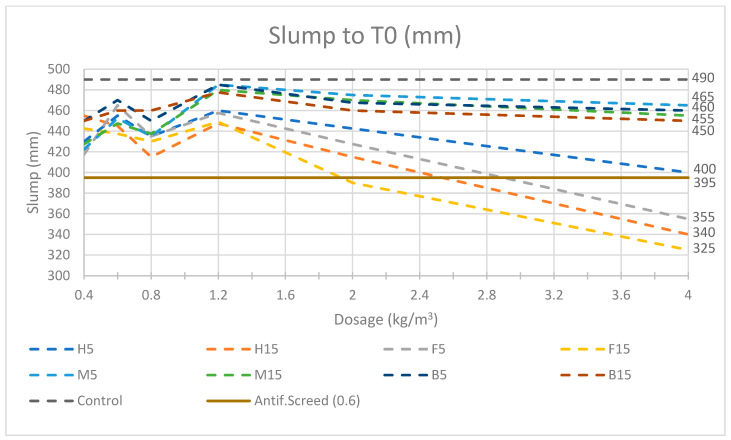
Graphique of slump at 0 min for soft and rigid fibers for all fiber dosages.

**Figure 8 materials-16-02203-f008:**
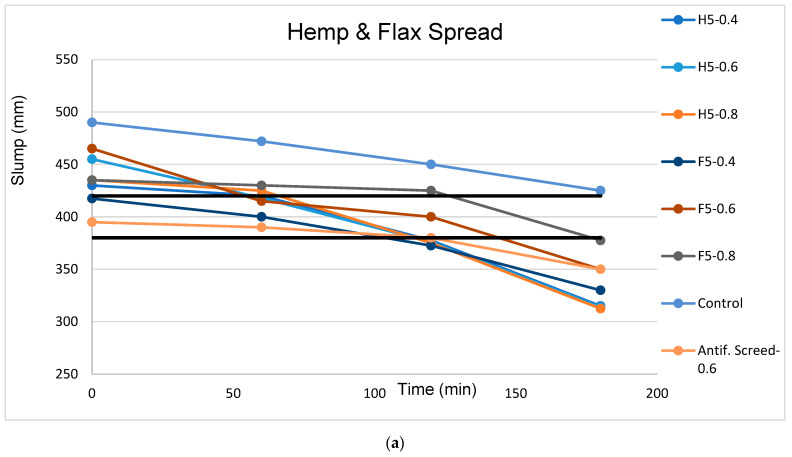
(**a**) Slump versus time for soft 5 mm fibers (**b**) Slump versus time for soft 15 mm fibers (**c**) Slump versus time for rigid 5 mm fibers (**d**) Slump versus time for rigid 15 mm fibers.

**Figure 9 materials-16-02203-f009:**
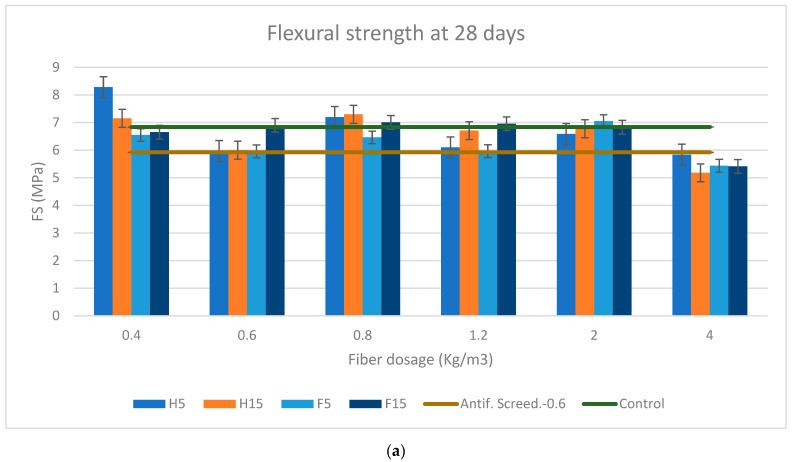
(**a**) Flexural strength at 28 days for soft fibers. (**b**) Flexural strength at 28 days for rigid fibers. (**c**) Flexural strength at 90 days for soft fibers. (**d**) Flexural strength at 90 days for rigid fibers.

**Figure 10 materials-16-02203-f010:**
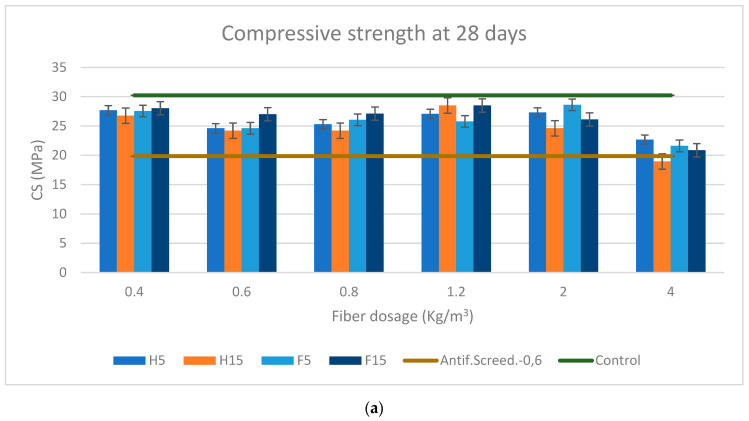
(**a**) Compressive strength at 28 days for soft fibers. (**b**) Compressive strength at 28 days for rigid fibers. (**c**) Compressive strength at 90 days for soft fibers. (**d**) Compressive strength at 90 days for rigid fibers.

**Figure 11 materials-16-02203-f011:**
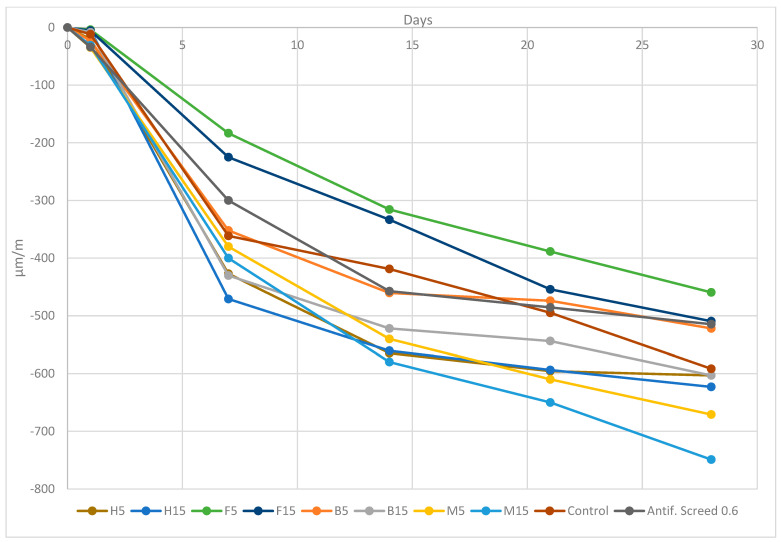
Drying shrinkage measured for FRS 2 kg/m^3^ of fiber dosage with vegetal, synthetic fibers and plain mortar at 1, 7, 14, 21 and 28 days.

**Figure 12 materials-16-02203-f012:**
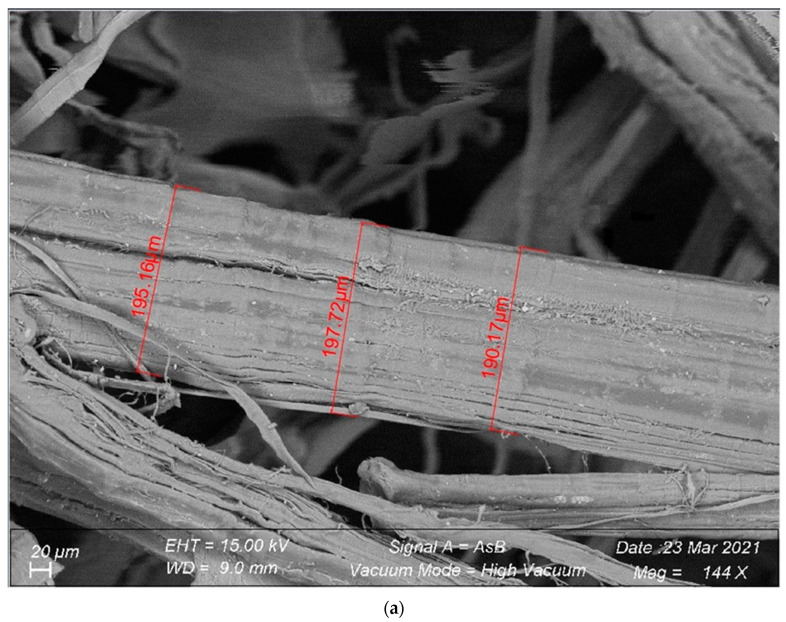
SEM images enabling an assessment of the distribution of the fibers and their relationships with the matrix taken by SEM in high vacuum mode: (**a**) Raw hemp fibers (× 144–Zoom × 20). (**b**) Hemp fibers in cement mortar (× 500–Zoom × 10). (**c**) Raw flax fibers (× 300–Zoom × 20). (**d**) Flax fibers in cement mortar (× 300–Zoom × 20). (**e**) Raw bamboo fibers (× 300–Zoom × 10). (**f**) Bamboo fibers in cement mortar (× 279–Zoom × 20). (**g**) Raw miscanthus fibers (× 140–Zoom × 30). (**h**) Miscanthus fibers in cement mortar (× 37–Zoom × 100).

**Table 1 materials-16-02203-t001:** Actual fiber lengths (average) as function of the fiber material and nominal length.

	Flax	Hemp	Miscanthus	Bamboo
***l_nom_* = 5 mm**	2	2	3.5	3.5
***l_nom_* = 15 mm**	4	4	7.5	7.5

**Table 2 materials-16-02203-t002:** Chemical composition of cement (%).

Components	SiO_2_	Al_2_O_3_	CaO	MgO	SO_3_	MnO	TiO_2_	Cl	K_2_O	SrO
**Proportions**	15.30	4.07	3.53	68.48	2.10	4.19	0.05	0.33	0.09	1.50

**Table 3 materials-16-02203-t003:** Adopted anti-fissuration screed mortar dosage.

Water/Binder Ratio	Strength Class	Flow Class	Equivalent Binder (EN 206)	Sand	Adjuvant
0.61	C20	S4	CEM II/A LL 42.5 R Limestone filler (382.5 kg/m^3^)	0/4 (1251 kg/m^3^)	Fluidifier (4.73 kg/m^3^)

**Table 4 materials-16-02203-t004:** Fiber content in kg/m^3^ and in percentage by volume.

kg/m^3^	0.4	0.6	0.8	1.2	2
**Fiber volume %**	0.03	0.04	0.06	0.08	0.14

**Table 5 materials-16-02203-t005:** Water added in the mix as a function of the fiber material and nominal length.

**Fiber dosage: 0.4 kg/m3**	**F5**	**F15**	**H5**	**H15**	**M5**	**M15**	**B5**	**B15**
**Water added (kg/m3)**	249.40	249.31	249.31	249.29	249.72	249.84	249.16	249.12
**Fiber dosage: 4 kg/m3**	**F5**	**F15**	**H5**	**H15**	**M5**	**M15**	**B5**	**B15**
**Water added (kg/m3)**	255.13	254.23	254.23	254.04	258.37	259.53	252.78	252.38

**Table 6 materials-16-02203-t006:** Aspect ratio of the fibers.

	Aspect Ratio of Raw Fibers (L/D)	Aspect Ratio of Fiber (L/D) Mixed with Mortar
H5	2/0.194 = 10	2/0.013 = 154
H15	4/0.194 = 20	4/0.013 = 308
F5	2/0.054 = 37	2/0.012 = 167
F15	4/0.054 = 74	4/0.012 = 333
M5	3.5/0.276 = 13	3.5/0.276 = 13
M15	7.5/0.276 = 27	7.5/0.276 = 27
B5	3.5/0.205 = 17	3.5/0.205 = 17
B15	7.5/0.205 = 37	7.5/0.205 = 37

**Table 7 materials-16-02203-t007:** Density (kg/m^3^) mean value measured on three specimens for each mortar mix. CS: Control sample—Volumic mass range respect CS; Volumic mass CS: 2245.5 kg/m^3^.

Mortar Mix	Dosage (kg/m^3^)											
	0.4	CS (%)	0.6	CS (%)	0.8	CS (%)	1.2	CS (%)	2	CS (%)	4	CS (%)
F5	2211.7	1.51	2213.7	1.42	2201.0	1.98	2198.4	2.10	2154.6	4.05	2005.2	10.70
F15	2214.7	1.37	2210.0	1.58	2211.8	1.50	2181.1	2.87	2141.8	4.62	1991.7	11.30
H5	2228.7	0.75	2221.7	1.06	2220.0	1.14	2160.9	3.77	2149.8	4.26	1993.1	11.24
H15	2219.8	1.14	2218.9	1.18	2220.0	1.14	2159.0	3.85	2147.9	4.35	1975.7	12.02
M5	2225.0	0.91	2230.0	0.69	2227.8	0.79	2225.3	0.90	2222.7	1.02	2193.9	2.30
M15	2227.3	0.81	2230.0	0.69	2228.4	0.76	2199.7	2.04	2180.0	2.92	2172.8	3.24
B5	2236.6	0.40	2235.8	0.43	2236.8	0.39	2205.5	1.78	2202.5	1.91	2189.7	2.48
B15	2231.2	0.64	2234.2	0.50	2228.8	0.74	2221.8	1.06	2221.5	1.07	2195.3	2.24

**Table 8 materials-16-02203-t008:** Air trapped (%) mean value measured on three specimens for each mortar mix. CS: Control sample—Air trapped range respect CS; Air trapped CS: 3%.

Mortar Mix	Dosage (kg/m^3^)											
	0.4	CS (%)	0.6	CS (%)	0.8	CS (%)	1.2	CS (%)	2	CS (%)	4	CS (%)
F5	3.60%	20.00	3.70%	23.33	4%	33.33	4.10%	36.67	4.40%	46.67	5.10%	70.00
F15	3.50%	16.67	3.40%	13.33	3.80%	26.67	3.60%	20.00	4.10%	36.67	4.80%	60.00
H5	3.70%	23.33	3.60%	20.00	3.70%	23.33	4.40%	46.67	4.50%	50.00	5.20%	73.33
H15	3.60%	20.00	3.50%	16.67	3.60%	20.00	4.10%	36.67	4.30%	43.33	5.40%	80.00
M5	3.40%	13.33	3.20%	6.67	3.10%	3.33	3.20%	6.67	3.40%	13.33	4.10%	36.67
M15	3.10%	3.33	3.20%	6.67	3.30%	10.00	3.50%	16.67	4.00%	33.33	4.60%	53.33
B5	3.20%	6.67	3.30%	10.00	3.30%	10.00	3.10%	3.33	3.40%	13.33	4.00%	33.33
B15	3.30%	10.00	3.20%	6.67	3.40%	13.33	3.20%	6.67	3.60%	20.00	4.20%	40.00

**Table 9 materials-16-02203-t009:** (**a**) Flexural strength (FS) at 28 days (MPa) mean value measured on three specimens for each mortar mix. (**b**) Flexural strength at 90 days (MPa) mean value measured on three specimens for each mortar mix. AS: antifissuration screed mortar (0.6 kg/m^3^ of fiber dosage)—Flexural strength range respect AS; Flexural strength AS: 5.92 MPa.

**(a)**												
**Mortar Mix**	**Dosage (kg/m^3^)**											
	**0.4**	**AS (%)**	**0.6**	**AS (%)**	**0.8**	**AS (%)**	**1.2**	**AS (%)**	**2**	**AS (%)**	**4**	**AS (%)**
F5	6.55	24.79	5.96	13.52	6.46	23.13	5.97	13.66	7.05	34.34	5.44	3.60
F15	7.27	38.61	6.50	23.84	7.01	33.63	6.96	32.55	6.83	30.16	5.41	3.16
H5	8.28	57.77	5.97	13.75	7.20	37.15	6.10	16.23	6.59	25.50	5.83	11.18
H15	7.15	36.29	6.00	14.39	7.30	39.04	6.71	27.83	6.78	29.10	5.18	−1.32
M5	6.56	24.92	6.66	26.87	6.77	28.92	7.37	40.35	7.57	44.30	6.27	19.56
M15	7.06	34.60	6.55	24.82	7.22	37.49	7.32	39.55	7.47	42.32	6.30	19.95
B5	6.89	31.30	6.83	30.19	6.47	23.29	7.34	39.95	7.89	50.33	6.84	30.28
B15	6.94	32.27	6.57	25.11	7.07	34.64	6.58	25.35	7.81	48.82	6.89	31.33
**(b)**												
**Mortar Mix**	**Dosage (kg/m^3^)**											
	**0.4**	**AS (%)**	**0.6**	**AS (%)**	**0.8**	**AS (%)**	**1.2**	**AS (%)**	**2**	**AS (%)**	**4**	**AS (%)**
F5	6.76	28.86	5.51	4.92	6.48	23.50	6.91	31.76	7.52	43.34	6.81	29.76
F15	6.58	25.36	7.20	37.18	6.33	20.62	7.97	51.79	7.18	36.84	6.65	26.71
H5	7.76	47.78	6.49	23.74	5.72	9.05	8.96	70.79	7.27	38.60	6.70	27.67
H15	6.30	20.07	6.08	15.83	5.37	2.34	8.19	56.15	6.87	30.88	6.02	14.71
M5	7.07	34.74	7.17	36.65	6.98	33.01	7.42	41.34	8.08	53.94	8.19	56.06
M15	6.82	29.91	7.45	41.94	7.13	35.80	7.97	51.82	8.23	56.87	8.18	55.87
B5	7.14	35.99	6.49	23.59	7.10	35.22	7.75	47.61	8.74	66.53	8.39	59.87
B15	6.95	32.37	7.28	38.69	6.81	29.68	7.73	47.22	8.66	65.01	8.50	61.97

**Table 10 materials-16-02203-t010:** (**a**) Compressive strength at 28 days (MPa) mean value measured on six specimens for each mortar mix. (**b**) Compressive strength at 90 days (MPa) mean value measured on six specimens for each mortar mix. AS: antifissuration screed mortar (0.6 kg/m^3^ of fiber dosage)—Compressive strength range respect AS; Compressive strength AS: 19.86 MPa.

**(a)**													
**Mortar mix**	**Dosage (kg/m^3^)**												
	**0.4**		**AS (%)**	**0.6**	**AS (%)**	**0.8**	**AS (%)**	**1.2**	**AS (%)**	**2**	**AS (%)**	**4**	**AS (%)**
F5	27.55		38.74	24.61	23.93	26.06	31.22	25.79	29.86	28.61	44.07	21.61	8.82
F15	28.03		41.16	27.01	36.00	27.11	36.54	28.50	43.53	26.12	31.53	20.86	5.04
H5	27.69		39.42	24.60	23.88	25.29	27.35	27.07	36.32	27.30	37.47	22.68	14.21
H15	26.76		34.73	24.20	21.85	24.20	21.87	28.50	43.52	24.61	23.91	18.95	−4.57
M5	26.62		34.03	27.05	36.20	27.72	39.59	32.35	62.89	31.55	58.86	25.82	30.02
M15	27.69		39.44	27.21	37.02	28.84	45.24	31.26	57.39	31.11	56.66	26.15	31.68
B5	32.14		61.82	27.87	40.36	30.05	51.31	32.88	65.59	31.26	57.41	29.81	50.12
B15	31.54		58.84	27.31	37.54	29.11	46.57	31.27	57.44	32.37	63.00	28.60	44.01
**(b)**													
**Mortar mix**		**Dosage (kg/m^3^)**											
		**0.4**	**AS (%)**	**0.6**	**AS (%)**	**0.8**	**AS (%)**	**1.2**	**AS (%)**	**2**	**AS (%)**	**4**	**AS (%)**
F5		31.70	46.50	26.46	22.27	29.92	38.26	26.85	24.08	28.95	33.77	23.36	7.96
F15		31.17	44.07	29.09	34.41	31.20	44.18	30.32	40.10	28.98	33.92	22.75	5.14
H5		30.72	41.95	27.03	24.92	27.16	25.52	34.61	59.95	27.54	27.27	23.27	7.54
H15		30.63	41.55	25.88	19.59	24.50	13.24	30.99	43.20	26.02	20.23	21.43	−0.96
M5		32.90	52.05	30.45	40.74	32.25	49.02	33.91	56.73	34.00	57.12	27.51	27.13
M15		32.07	48.20	30.96	43.10	31.04	43.43	33.10	52.95	33.45	54.59	29.17	34.81
B5		32.69	51.08	29.85	37.95	33.06	52.77	36.06	66.65	33.69	55.71	31.67	46.36
B15		34.72	60.45	30.64	41.59	31.91	47.46	34.14	57.78	34.00	57.14	29.10	34.48

## Data Availability

All data is published in this article.

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
