# Peer review of "Valorization of Vegetal Fibers (Hemp, Flax, Miscanthus and Bamboo) in a Fiber Reinforced Screed (FRS) Formulation"

_materials, 2023, doi:10.3390/ma16062203_

Round 1

Reviewer 1 Report

The authors investigated the influence of fibre content and type to the physical and mechanical properties of screed mortar, especially for vegetal fibers, considering. Considering the environment and ecological problems, this is an interesting topic. However, there are several problems that need to be checked.

1. In table 1, please give the difference between the average length and the nominal length.

2. Figure 4 is not clear.

3. For the flexural compressive strength tests, the authors didnt give the size of the specimens.

4. Some comparison with the results reported by other authors can be discussed.

Author Response

Thank you for your effort in allowing us to improve the article with your comments.
We remain at your disposal for any further improvement or modification of the article. We hope that the modifications made will be to your appreciation.

Best regards,

Sergio Pons Ribera

Reviewer 2 Report

The authors present a well-written paper related to the use of vegetal soft and rigid fibers in a screed-mortar to have a sustainable fabric-cement matrix. The topic is interesting and coherent with the journal's scope, however, the manuscript should be improved before potential publication. Below you can find some suggestions:

1. Please avoid lumping references (such as "numerous investigations to develop sustainable building materials using natural fibers [22-32”). Where it is possible, each reference should be described separately.

2. Please mention the Graphical abstract.

3. Please clearly point out the main novelties of your work - what is your contribution to the actual state of the art? Such a part should be placed at the end of the "Introduction" section.

4. The quality of the images/figures should be improved.

5. Please avoid repetitions of the information presented, such as Figure 5 and Table 6 or Figure 6 and Table 7. I think that the information presented is the same.

6. Please improve the Conclusions section and the discussion of the experimental results obtained. Please compare your results with other results presented in literature.

Author Response

(The authors gave the same response as above.)

Author Response

(The authors gave the same response as above.)
